# Microbe-Derived Antioxidants Protect IPEC-1 Cells from H_2_O_2_-Induced Oxidative Stress, Inflammation and Tight Junction Protein Disruption via Activating the Nrf2 Pathway to Inhibit the ROS/NLRP3/IL-1β Signaling Pathway

**DOI:** 10.3390/antiox13050533

**Published:** 2024-04-27

**Authors:** Cheng Shen, Zhen Luo, Sheng Ma, Chengbing Yu, Ting Lai, Shangshang Tang, Hongcai Zhang, Jing Zhang, Weina Xu, Jianxiong Xu

**Affiliations:** Shanghai Key Laboratory of Veterinary Biotechnology/Shanghai Collaborative Innovation Center of Agri-Seeds, School of Agriculture and Biology, Shanghai Jiao Tong University, Shanghai 200240, China; shencheng1233@sjtu.edu.cn (C.S.); luozhen0615@sjtu.edu.cn (Z.L.); 1329109669@sjtu.edu.cn (S.M.); lai-ting@sjtu.edu.cn (T.L.); shang-t@sjtu.edu.cn (S.T.); hczhang@sjtu.edu.cn (H.Z.); zhangjing224@sjtu.edu.cn (J.Z.); xuweina@sjtu.edu.cn (W.X.)

**Keywords:** microbe-derived antioxidants, oxidative stress, inflammatory response, tight junction proteins, Nrf2/ROS/NLRP3/IL-1β

## Abstract

Oxidative stress can induce inflammation and tight junction disruption in enterocytes. The initiation of inflammation is thought to commence with the activation of the ROS/NLRP3/IL-1β signaling pathway, marking a crucial starting point in the process. In our previous studies, we found that microbe-derived antioxidants (MAs) showed significant potential in enhancing both antioxidant capabilities and anti-inflammatory effects. The main aim of this research was to investigate the ability of MAs to protect cells from oxidative stress caused by H_2_O_2_, to reduce inflammatory responses, and to maintain the integrity of tight junction proteins by modulating the ROS/NLRP3/IL-1β signaling pathway. IPEC-1 cells (1 × 10^4^ cells/well) were initially exposed to 100 mg/L of MAs for 12 h, after which they were subjected to 1 mM H_2_O_2_ treatment for 1 h. We utilized small interfering RNA (siRNA) to inhibit the expression of NLRP3 and Nrf2. Inflammatory factors such as IL-1β and antioxidant enzyme activity levels were detected by ELISA. Oxidative stress marker ROS was examined by fluorescence analysis. The NLRP3/IL-1β signaling pathway, Nrf2/HO-1 signaling pathway and tight junction proteins (ZO-1 and Occludin) were detected by RT-qPCR or Western blotting. In our research, it was observed that MA treatment effectively suppressed the notable increase in H_2_O_2_-induced inflammatory markers (TNF-α, IL-1β, and IL-18), decreased ROS accumulation, mitigated the expression of NLRP3, ASC, and caspase-1, and promoted the expression of ZO-1 and Occludin. After silencing the NLRP3 gene with siRNA, the protective influence of MAs was observed to be linked with the NLRP3 inflammasome. Additional investigations demonstrated that the treatment with MAs triggered the activation of Nrf2, facilitating its translocation into the nucleus. This process resulted in a notable upregulation of Nrf2, NQO1, and HO-1 expression, along with the initiation of the Nrf2-HO-1 signaling pathway. Consequently, there was an enhancement in the activities of antioxidant enzymes like SOD, GSH-Px, and CAT, which effectively mitigated the accumulation of ROS, thereby ameliorating the oxidative stress state. The antioxidant effectiveness of MAs was additionally heightened in the presence of SFN, an activator of Nrf2. The antioxidant and anti-inflammatory functions of MAs and their role in regulating intestinal epithelial tight junction protein disruption were significantly affected after siRNA knockdown of the Nrf2 gene. These findings suggest that MAs have the potential to reduce H_2_O_2_-triggered oxidative stress, inflammation, and disruption of intestinal epithelial tight junction proteins in IPEC-1 cells. This reduction is achieved by blocking the ROS/NLRP3/IL-1β signaling pathway through the activation of the Nrf2 pathway.

## 1. Introduction

Inflammation serves as the body’s protective reaction aimed at removing detrimental stimuli. It activates in response to various harmful triggers such as damaged cells, invading pathogens, or endotoxins. This process is integral to the innate immune system’s response [1]. Nevertheless, an overabundance of inflammatory reactions can inflict harm on bodily tissues. Due to its status as the primary immune organ, the gut is particularly prone to experiencing such repercussions. Excessive inflammation may precipitate various chronic gut-related ailments characterized by inflammation, including Crohn’s disease and ulcerative colitis [2,3]. The correlation between inflammation and oxidative stress, along with heightened levels of intracellular ROS, is widely acknowledged, with the latter being recognized as among the most influential instigators of inflammation [4]. For example, excessive production of reactive oxygen species (ROS) not only promotes the secretion of inflammatory cytokines, but excessive production of these cytokines can lead to disruption of intestinal integrity and epithelial function [5,6]. Overproduction of ROS also provides the necessary signal for activation of the nucleotide-binding domain (NOD)-like receptor protein 3 (NLRP3) inflammasome [7]. Recent studies have linked NLRP3 inflammasome signaling to inflammatory bowel disease [8].

The nuclear factor erythroid-associated factor-2 (Nrf2) plays a pivotal role in cellular homeostasis by overseeing the expression of antioxidant and anti-inflammatory molecules [9,10]. After activation, Nrf2 detaches from kelch-like ECH-associated protein 1 (Keap1), relocates to the nucleus, and binds to antioxidant response elements (AREs) to control the expression of antioxidant genes like NAD(P) quinone oxidoreductase 1 (NQO-1) and heme oxygenase 1 (HO-1) [11]. There is growing evidence indicating that the activation of Nrf2 serves to mitigate the excessive generation of ROS, consequently impeding the activation of NLRP3 inflammasomes. Targeting Nrf2 signaling has emerged as a promising strategy for the management of inflammatory damage [12,13].

Microbe-derived antioxidants (MAs) are a mixture of sea buckthorn and prickly pear made by probiotic bacteria through solid–liquid complex fermentation, followed by extraction, concentration, inactivation, lyophilization and other processing techniques [14]. The UHPLC–QTOF-MS analysis revealed that MAs contain 426 metabolites, mainly including organic acids and derivatives (15.96%), organic oxygen compounds (10.80%), organoheterocyclic compounds (7.04%), lipids and lipid-like molecules (6.10%), benzenoids (4.46%), and phenylpropanoids and polyketides (4.23%) [15]. In our prior research, it was documented that MAs exhibit a commendable ability to scavenge free radicals [15], enhance hepatic antioxidant capacity in mothers and offspring [16], and regulate metabolic inflammation in mice [17], but whether MAs can mitigate oxidative stress and inflammatory response-induced disruption of intestinal epithelial tight junction through the Nrf2-ROS-NLRP3 signaling pathway is unclear. In this study, we employed IPEC-1 cells as a cellular model and developed Nrf2 and NLRP3 knockdown IPEC-1 cells to explore the molecular pathways involved in the impact of MAs on oxidative stress, inflammatory response, and disruption of tight junction proteins.

## 2. Materials and Methods

### 2.1. Reagents

The chemicals utilized in this investigation were acquired from the following suppliers: Malondialdehyde (MDA, A003-1-2), glutathione peroxidase (GSH-Px, A005-1-2), catalase (CAT, A007-1-1) and Superoxide dismutase (SOD, A001-3-2) ELISA kits were provided by Nanjing Jiancheng Institute of Biological Engineering (Nanjing, China). Nucleoprotein and cytoplasmic protein extraction kit (P0028), reactive oxygen species assay kit (S0033M), cellular CCK-8 kit (C0039), BCA protein concentration assay kit (P0010), and 2′, 7′-dichlorofluorescein diacetate (DCFH-DA, S0033) were sourced from Beyotime Biotechnology (Shanghai, China). RPMI 1640 medium (11875093), Fetal bovine serum (FBS, 10091148), 0.25% trypsin solution (25095-019) and penicillin streptomycin (15140122) were purchased from GIBCO (Carlsbad, CA, USA). Lipofectamine 2000 (12566014) was procured from Invitrogen (Carlsbad, CA, USA). RNA extraction kit I was acquired from OMEGA (R6841-01; Norcross, GA, USA). ECL assay (RPN2232) was purchased from GE Healthcare (Boston, USA). PrimeScrip RT kit was provided by TaKaRa (RR047A; Ōsaka, Japan). PVDF (No. IPVH00010) was sourced from Millipore (Boston, MA, USA). Primary antibodies against NLRP3 (NBP1-31245), ZO-1 (NBP1-85047) and IL-1β (NB600-633) were purchased from Novus (Saint Louis, MI, USA). Occludin (LS-B5737-100) and ASC (LS-C40344-100) were sourced from Lifespan (Saint Louis, MI, USA). Lamin B (abs131244) and HO-1 (abs123962) were purchased from Absin (Shanghai, China). NQO1 (ab2346) and Anti-rabbit IgG HRP antibody (ab97051) were obtained from Abcam (Cambs, UK). Nrf2 (A2019-100) was acquired from Biovision (San Francisco, CA, USA). IL-18 (AF588) was provided by R&D Systems (Minneapolis, MN, USA). Caspase-1 (sc-56036) was procured from Santacruz (Santa Cruz, CA, USA). GAPDH (5174) was provided by CST (Boston, MA, USA). MA (Trade name KB-120) was provided by Jianghan Biotechnology (Shanghai) Co., Ltd. (Shanghai, China), with its main ingredients as described by Luo et al. [15].

### 2.2. Cell Culture and Processing

IPEC-1 cells, obtained from Shanghai Yaji Biotechnology Co. (Shanghai, China), were cultured in 1640 medium complemented with 10% (*v*/*v*) FBS, 100 mg/mL streptomycin, and 100 U/mL penicillin. The cultures were kept in a humidified environment at 37 °C with 5% CO_2_.

### 2.3. Cell Viability Assay

IPEC-1 cells were plated in 96-well plates at a density of 1 × 10^4^ cells per well and cultured at 37 °C for 24 h. MAs were dissolved in a complete medium, vigorously mixed, and subsequently passed through a 0.45 μm Merck Millipore end filter for preparation prior to use. To assess the impact of various concentrations of MAs on cell viability, when the cells were cultured to 70–80% confluency, they were exposed to varying concentrations of MAs (0, 10, 20, 50, 100, and 200 mg/L) for a period of 12 h. The liquid above was discarded, and the sample was washed three times with DPBS. Subsequently, it was exposed to 1 mM H_2_O_2_ (dissolved in a complete medium) for 1 h. The treated cells were discarded from the supernatant, washed three times with DPBS. Then, 100 μL of medium and 10 μL of CCK-8 were added and the mixture was incubated at 37 °C for 2 h. The optical density was assessed at 490 nm, and the results were reported as a fold change relative to the untreated group.

### 2.4. Measurement of Intracellular ROS

To assess the impact of MAs on intracellular ROS levels, the cells underwent initial treatment with 100 mg/L MAs for a duration of 12 h. The supernatant was removed, and the sample was washed three times with DPBS. Subsequently, it was subjected to treatment with 1 mM H_2_O_2_ (dissolved in a complete medium) for 1 h. The cells were subsequently rinsed twice using DPBS. Following the treatment, the cells were subjected to a 20 min incubation at 37 °C in the presence of a 10 μM DCFH-DA probe. The cells were subsequently rinsed twice using DPBS. Fluorescence readings were taken using a fluorescent microplate reader with excitation/emission wavelengths of 535/610 nm. Fluorescence images were captured using fluorescence microscopy (Tokyo, Japan).

### 2.5. Determination of Antioxidant Enzyme Activity

The levels of MDA and the activities of SOD, CAT, and GSH-Px were assessed utilizing the detection kits (Nanjing Jiancheng Bioengineering Institute, Nanjing, China) in accordance with the guidelines provided by the manufacturer. The measurement of protein concentrations was conducted using the Bradford Protein Assay Kit (Shanghai, China). The absorbance at 550 nm for the SOD and GSH-Px measurements and the absorbances at 532 nm and 405 nm for the MDA and CAT measurements, respectively, were measured using a 96-well plate reader (Synergy 2, BioTek, Winooski, VT, USA).

### 2.6. Inflammatory Cytokine Detection

Levels of TNF-α, IL-1β, IL-6 and IL-18 bioactivity in cell cultures were assayed by the corresponding porcine ELISA kits according to the protocol. Briefly, the plates were coated with the corresponding antibodies. This was followed by detection with a horseradish peroxidase-labeled substrate after incubation for 10 min at 37 °C. The plates were then read in a 96-well plate reader (Synergy 2, BioTek, USA) at 450 nm.

### 2.7. Reverese Transcription-Quantitative Real-Time Polymerase Chain Reaction (RT-qPCR)

The qPCR analysis was performed as described previously [15]. Total RNA from the treated cells was extracted using an RNA extraction kit and the RNA was reverse transcribed into cDNA using the PrimeScrip RT kit. The primers used in this investigation were designed in accordance with prior research findings and are provided in Appendix A. β-actin was utilized as the reference gene to standardize the transcription of the target gene. Relative gene expression was assessed utilizing the 2^−∆∆Ct^ methodology.

### 2.8. Western Blot Analysis

Following the treatment of cells, the protein concentrations within the whole cell extracts were assessed in accordance with the guidelines outlined in the BCA Protein Assay Kit. Equivalent quantities of total protein (20–40 μg) were loaded onto SDS-polyacrylamide gel for electrophoresis, and subsequently transferred onto PVDF membranes. The membranes were sealed by applying a solution comprising 5% skim milk powder in Tris-buffered saline supplemented with 0.05% Tween (TBST). After being washed with TBST, the membranes were left to incubate overnight at 4 °C with the primary antibodies. These antibodies included NLRP3 (1000), ASC (1:1000), Caspase-1 (1:1000), IL-1β (1:1000), IL-18 (1:1000), ZO-1 (1:1000), Occludin (1:1000), Nrf2 (1:500), NQO1 (1:1000), HO-1 (1:1000), Lamin B (1:1000), and GAPDH (1:1000). The detection of secondary antibody binding involved the use of an anti-rabbit antibody (1:2000), followed by visualization using ECL detection reagents. We utilized an advanced chemiluminescence detection system (Tanon, Shanghai, China) for capturing images. The density of the protein bands was quantified utilizing Image J V2.6 software.

### 2.9. RNA Silencing

Cells underwent transfection using negative control NC siRNA, NLRP3 siRNA, or Nrf2 siRNA, employing tailored siRNA reagent systems as per the manufacturer’s guidelines. The cells were collected at 48 h for mRNA or protein expression analysis to evaluate the effectiveness of silencing, or they were treated with their respective agents and subjected to subsequent treatments as described earlier. The primer sequences utilized in this investigation are detailed in Appendix A. After transfection, the cells underwent a 12 h pre-treatment with the designated concentration of MAs, after which they were either stimulated with H_2_O_2_ (1 mM) for 1 h or left untreated. The cells were processed, and the levels of NLRP3 and Nrf2 were assessed using real-time PCR and Western blot analysis.

### 2.10. Statistical Analysis

The statistical significance of mean value differences was assessed through one-way analysis of variance, subsequently followed by Duncan’s multiple range tests for comparison; other data were analyzed using Student’s *t*-test. The presented data are shown as the mean ± standard error of the mean (SEM) derived from a minimum of three replicates, and represented as mean ± SEM. Statistical significance analysis was conducted using SPSS 17.0 software (SPSS Inc., Chicago, IL, USA). Results with *p*-values below 0.05 were deemed statistically significant.

## 3. Results

### 3.1. MAs Demonstrate Anti-Inflammatory Properties in H_2_O_2_-Induced IPEC-1 Cells

When IPEC-1 cells were subjected to 100 and 200 mg/L of MA treatment, their cell viability exhibited a notable increase in comparison to the control group. Additionally, IPEC-1 cells demonstrated significant protection against H_2_O_2_-induced cytotoxicity under these conditions (Figure 1A). No notable disparity in the improvement in cell viability was observed between 100 and 200 mg/L MA. Consequently, subsequent experiments were conducted utilizing 100 mg/L of MAs. MA pre-treatment was found to have no significant effect on the mRNA expression of NLRP3, IL-1β, ZO-1, and Occludin compared to the control group (Appendix A). The induction of H_2_O_2_ significantly elevated the mRNA levels of TNF-α, IL-1β, IL-6, and IL-18. However, MAs notably inhibited this elevation, exerting a significant inhibitory effect (Figure 1B). Additionally, we noted a rise in IL-1β and IL-18 protein levels following H_2_O_2_ induction. Conversely, the protein levels of IL-1β and IL-18 were notably diminished when cells were co-treated with MAs (Figure 1C). The levels of inflammatory factors, such as TNF-α, IL-1β, IL-6, and IL-18, significantly increased following exposure to H_2_O_2_ induction, which was notably suppressed by MA treatment (Figure 1D).

### 3.2. MAs Mitigate the Inflammatory Effects Induced by H_2_O_2_ by Inhibiting the NLRP3 Inflammasome

The gene expression levels and protein expression levels of NLRP3, ASC and Caspase-1 exhibited significant increments following H_2_O_2_ induction. However, this increase was markedly suppressed by MA treatment (Figure 2A,B). To further confirm the involvement of the NLRP3 inflammasome in the NLRP3/IL-1β signaling pathway induced by H_2_O_2_, RNA silencing experiments targeting NLRP3 were conducted. After transfection with NLRP3 siRNA, approximately 77% of NLRP3 mRNA level expression was significantly reduced (Figure 2C). Transfecting with NLRP3 siRNA notably decreased the elevation of IL-1β and IL-18 mRNA levels induced by H_2_O_2_ stimulation. Moreover, this decrease was significantly augmented by pre-treatment with MAs, demonstrating synergistic effects (Figure 2D). Transfecting with NLRP3 siRNA led to a notable decrease in the elevated mRNA levels of NLRP3, ASC, and Caspase-1 induced by H_2_O_2_ stimulation. Moreover, this decrease was notably augmented with the concurrent administration of MAs (Figure 2E). In response to H_2_O_2_ stimulation, IL-1β exhibited a reduction of 32.1% after the transfection of NLRP3 siRNA and a further reduction of 53.6% upon the addition of MAs (Figure 2F).

### 3.3. MAs Attenuate H_2_O_2_-Induced Tight Junction Protein Disruption by Inhibiting the NLRP3 Inflammasome

The expression levels of ZO-1 and Occludin genes, as well as their protein expression, experienced a notable decrease upon exposure to H_2_O_2_, which was substantially mitigated by MA treatment (Figure 3A,B). Transfection of NLRP3 siRNA notably mitigated the reduction in mRNA and protein expression levels of ZO-1 and Occludin in H_2_O_2_-stimulated IPEC-1 cells. Moreover, this mitigating effect was significantly amplified by pre-treatment with MAs (Figure 3C,D).

### 3.4. MAs Mitigate the Oxidative Stress Induced by H_2_O_2_ in IPEC-1 Cells

Cells subjected to 1 mM H_2_O_2_ for 1 h displayed over a four-fold rise in intracellular ROS levels. Conversely, administration of 100 mg/L MA significantly suppressed the heightened ROS levels. Moreover, co-administration with an Nrf2 activator (SFN) further enhanced the inhibition of ROS production (Figure 4A). The SOD, GSH-Px, and CAT activities experienced a swift decline upon exposure to H_2_O_2_, while the MDA content surged, signaling a rapid reduction in cellular antioxidant capacity. This decline was mitigated by MAs and further bolstered when combined with an Nrf2 activator (Figure 4B).

### 3.5. MAs Alleviate H_2_O_2_-Induced Cellular Inflammation via the Nrf2/NLRP3/IL-1β Signaling Pathway

Additional noteworthy decreases in both NLRP3 and IL-1β mRNA levels as well as protein expression were noted in IPEC-1 cells exposed to MAs in conjunction with SFN (an Nrf2 activator) when compared to cells treated solely with MAs (Figure 5A,B). The fluctuations in IL-1β levels followed a comparable pattern (Figure 5C). Treatment with MAs resulted in a notable elevation in Nrf2 protein levels within the nucleus (Figure 5D), consequently inducing a marked upsurge in both mRNA levels and protein expression of Nrf2, HO-1, and NQO1 in IPEC-1 cells (Figure 5E,F). This effect was further intensified when the Nrf2 activator SFN was present. To confirm the involvement of Nrf2 in the NLRP3/IL-1β signaling pathway induced by H_2_O_2_, experiments were conducted to silence Nrf2. After the cells were transfected with Nrf2 siRNA, there was a significant reduction of approximately 67% in Nrf2 mRNA expression and about 85% in Nrf2 protein expression (Figure 6A,B). Unlike the outcomes observed with the Nrf2 activator SFN, the suppression of the inflammatory response through the Nrf2/NLRP3/IL-1β signaling pathway by MAs was notably impeded following Nrf2 silencing (Figure 6C–G). When Nrf2 was suppressed, the capacity of MAs to enhance cellular antioxidant enzyme activity and decrease IL-1β secretion was notably impaired, leading to a significant rise in ROS levels and triggering severe oxidative stress and inflammatory reactions in the cells (Figure 6E,H,I).

### 3.6. MAs Attenuate H_2_O_2_-Induced Tight Junction Protein Disruption by Activating the Nrf2/NLRP3/IL-1β Signaling Pathway

Under H_2_O_2_ induction, there was a significant decrease observed in the gene expression levels and protein expression levels of ZO-1 and Occludin. However, concurrent treatment with the Nrf2 activator SFN significantly augmented the beneficial impact of MAs in mitigating the functional impairment associated with disruption of intestinal epithelial tight junction proteins (Figure 7A,B). Nrf2 siRNA transfection significantly inhibited the effect of MAs on the H_2_O_2_-stimulated increase in ZO-1 and Occludin mRNA as well as protein expression levels in an upward trend (Figure 7C,D).

### 3.7. MAs Attenuate H_2_O_2_-Induced Tight Junction Protein Disruption by Reducing IL-1β

IL-1β was able to cause a significant decrease in Occludin protein levels, and the presence of the IL-1β antagonist IL-1β AB notably enhanced the levels of the connexin Occludin (Figure 8A,B). When the IL-1β antagonist IL-1β AB was co-treated with H_2_O_2_, it not only significantly suppressed the significant decrease in Occludin levels caused by H_2_O_2_ treatment, but also mitigated the marked elevation in MLCK levels (Figure 8C).

## 4. Discussion

The intestinal barrier serves as the primary defense mechanism of the body, shielding it from harmful antigens and pathogens present in the intestinal lumen. The intestinal epithelial cells create a barrier between the hostile external and internal environments. However, multiple factors such as infection, inflammation and oxidative stress can lead to intestinal epithelial cell damage and dysfunction [18]. Our research uncovered that upon stimulation with H_2_O_2_, IPEC-1 cells generated significant levels of ROS, subsequently triggering the activation of the NLRP3 signaling pathway. This activation resulted in the production of the IL-1β inflammatory factor and disruption of intestinal tight junction proteins. The treatment with MAs resulted in a notable decrease in ROS levels, suppressed the activation of the NLRP3 inflammasome, thereby decreasing the IL-1β inflammatory factor, and mitigated disruption in intestinal tight junction proteins. In this study, we illustrate that MAs exhibit a protective effect on cells against H_2_O_2_-induced cellular inflammation and tight junction protein disruption. Additional research revealed that the defensive mechanism exerted by MAs was linked to the initiation of Nrf2 antioxidant signaling. MAs triggered the relocation of Nrf2 to the nucleus, consequently augmenting the efficacy of numerous antioxidant enzymes. This protection is achieved by suppressing the ROS-NLRP3-IL-1β signaling axis through activation of the Nrf2-antioxidant signaling pathway. The present research demonstrates that MAs effectively hinder the onset stage of cellular inflammation induced by H_2_O_2_, indicating their promising capability to safeguard against intestinal tight junction protein disruption by early intervention and alleviation of inflammation.

The inflammatory response is closely associated with oxidative stress. Several investigations have indicated the necessity of ROS for triggering the activation of the NLRP3 inflammasome. Additionally, certain studies have suggested that ROS play a role in influencing the initial phase of NLRP3 inflammasome activation [19,20]. Upon activation triggered by stimuli such as ROS, NLRP3 initiates the recruitment of the bridging protein ASC. Subsequently, ASC facilitates the recruitment and activation of procaspase-1, ultimately resulting in the generation and maturation of IL-1β [21]. IL-1β has a dual role: it can trigger additional inflammatory pathways and also kickstart macrophages through the activation of IL-1R [22,23]. Increasing evidence suggests that by inhibiting NLRP3 activation, the production of IL-1β, a key component of the inflammatory response, can be reduced, thereby suppressing inflammation. Moreover, it is believed that intestinal barrier dysfunction correlates with the onset of intestinal inflammation, with various studies emphasizing the crucial role played by NLRP3 in maintaining the integrity of the intestinal barrier [24,25,26]. Inflammatory factors such as IL-1β cause an increase in intestinal tight junction permeability through pathways such as activation of the MLCK gene [23]. Our findings indicate that upon cell stimulation with H_2_O_2_ leading to a substantial production of ROS, there was a notable increase in both mRNA and protein levels of NLRP3, ASC, and caspase-1. This activation of the NLRP3 signaling pathway resulted in a significant elevation in mRNA and protein levels of IL-18 and IL-1β, a marked increase in extracellularly secreted IL-1β levels, along with a significant reduction in ZO-1 and Occludin. But all of these were subsequently reversed by treatment with MAs. Furthermore, the findings from the NLRP3 gene silencing experiments indicate that the NLRP3 inflammasome serves as a pivotal component in the inflammatory cascade. Moreover, it appears that MA can mitigate the secretion of inflammatory mediators, such as IL-1β, by inhibiting NLRP3 levels. The trend of significant decreases in ZO-1 and Occludin due to H_2_O_2_ stimulation was also suppressed after NLRP3 gene silencing, suggesting that the NLRP3 inflammasome plays a pivotal role in regulating intestinal barrier damage caused by inflammatory responses. IL-1β inflammatory factor may significantly reduce Occludin expression through activation of MLCK, leading to impairment of intestinal barrier function. These findings indicate that MAs might hinder the onset and activation of NLRP3 induced by H_2_O_2_ by diminishing ROS levels, thus alleviating oxidative stress. Additionally, it could mitigate inflammatory reactions and maintain the integrity of intestinal epithelial tight junction protein through modulation of the ROS-NLRP3-IL-1β signaling pathway.

Nrf2 plays a pivotal role in mitigating oxidative stress, inflammatory reactions, and barrier impairment, making it a prime target for ROS inhibition. Targeting Nrf2 signaling presents an efficacious strategy for treating colitis and intestinal barrier damage [27]. In our current investigation, we have shown that MAs notably enhance the translocation of Nrf2 to the nucleus. This, in turn, leads to the upregulation of various antioxidant enzyme activities via the Nrf2-HO-1 signaling pathway. Additionally, MAs diminish MDA content, lower ROS levels, suppress the activation of the NLRP3 inflammasome, and mitigate oxidative stress, inflammatory reactions and cellular barrier damage. MAs’ effectiveness was further amplified when combined with the Nrf2 activator SFN, yet was notably subdued upon Nrf2 silencing.

MAs encompass an array of antioxidant compounds with robust antioxidant properties [14]. In in vitro experiments, the MA blend not only has a scavenging capacity comparable to that of Vc for oxidants such as OH-, but also has an even higher scavenging capacity for superoxide than MitoQ [15]. In this study, it was observed that MAs provide cellular protection against H_2_O_2_-induced inflammation and maintain tight junction protein integrity by suppressing the ROS-NLRP3-IL-1β signaling pathway via activation of the Nrf2-antioxidant signaling pathway. This may be related to the presence of oleanolic acid, rutin, kaempferol and quercetin [15]. Oleanolic acid mitigates oxidative stress and diminishes neuronal apoptosis through the activation of the Nrf2/HO-1 pathway [28]. Rutin-altered selenium nanoparticles mitigate cellular oxidative damage triggered by H_2_O_2_ through the activation of the Nrf2/HO-1 signaling pathway [29]. Kaempferol prevents inflammation in the lungs via Nrf2/HO-1 [30]. Quercetin exerts anti-allergic, antioxidant, anti-inflammatory and anti-fibrotic activities via Nrf2/HO-1 [31].

## 5. Conclusions

In conclusion, we show here for the first time that MAs can attenuate H_2_O_2_-induced oxidative stress, inflammatory responses and intestinal epithelial tight junction protein disruption in IPEC-1 cells by inhibiting the ROS/NLRP3/IL-1β signaling pathway through activation of the Nrf2 pathway (Figure 9). This study illustrates MAs’ capacity to regulate inflammatory reactions and safeguard against intestinal tight junction disruption. Consequently, studies of MAs may provide ideas for the prevention of inflammatory bowel disease.

## Figures and Tables

**Figure 1 antioxidants-13-00533-f001:**
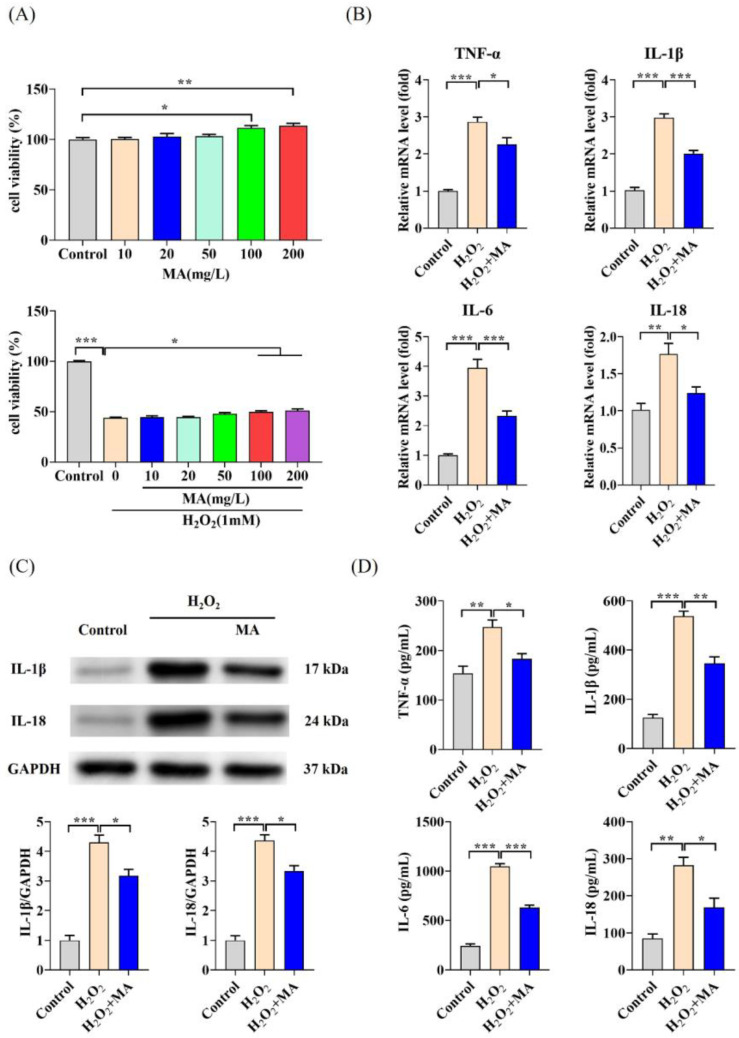
MAs demonstrate anti-inflammatory properties in H_2_O_2_-induced IPEC-1 cells. (**A**) The viability of IPEC-1 cells was determined using the CCK-8 assay. (**B**) The gene expression levels of TNF-α, IL-1β, IL-6, and IL-18 in IPEC-1 cells after treatment with 100 mg/L of MAs for 12 h, followed by stimulation with 1 mM H_2_O_2_ for 1 h. (**C**) The protein expression levels of IL-1β and IL-18. (**D**) The protein levels of TNF-α, IL-1β, IL-6, and IL-18. ** p* < 0.05, *** p* < 0.01, **** p* < 0.001.

**Figure 2 antioxidants-13-00533-f002:**
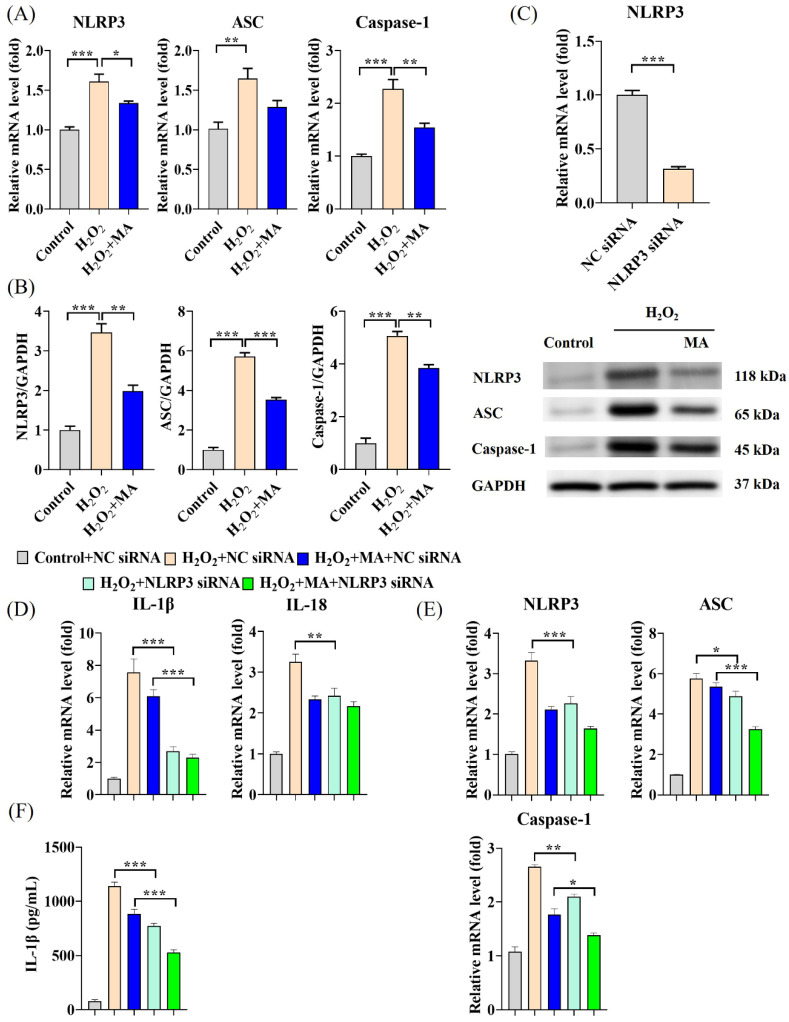
MAs mitigate the inflammatory effects induced by H_2_O_2_ by inhibiting the NLRP3 inflammasome. (**A**) The gene expression levels of NLRP3, ASC and Caspase-1. (**B**) The protein expression levels of NLRP3, ASC, and Caspase-1. (**C**) The gene expression level of NLRP3 after siRNA treatment. The cells underwent a sequential process starting with transfection with siRNA for 48 h, followed by exposure to 100 mg/L of MAs for 12 h, and concluding with treatment with 1 mM H_2_O_2_ for 1 h. All subsequent experimental cell treatments depicted in this figure follow the same procedure as this particular cell treatment. (**D**) The gene expression levels of IL-1β and IL-18 after siRNA treatment. (**E**) The gene expression levels of NLRP3, ASC, and Caspase-1 after siRNA treatment. (**F**) The protein level of IL-1β after siRNA treatment. ** p* < 0.05, *** p* < 0.01, **** p* < 0.001.

**Figure 3 antioxidants-13-00533-f003:**
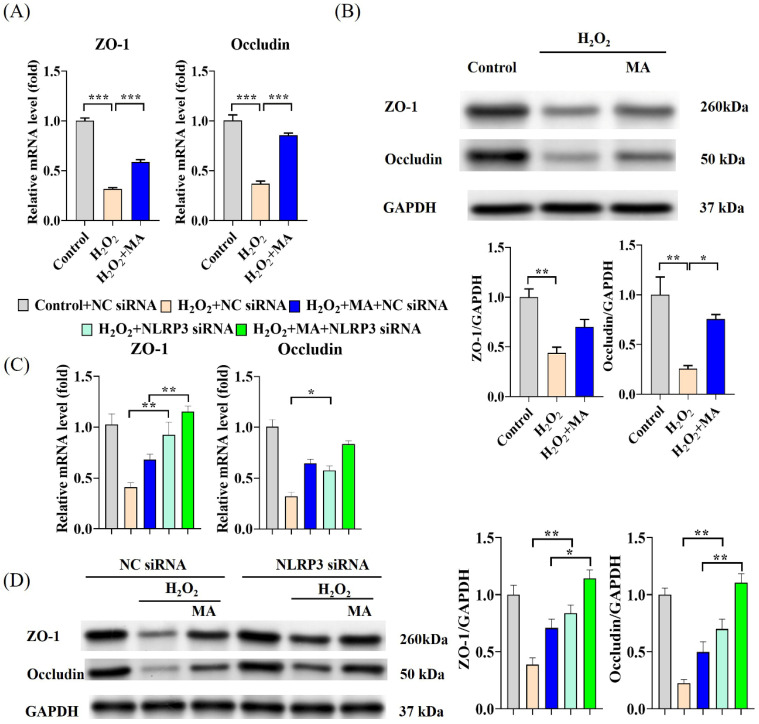
MAs attenuate H_2_O_2_-induced barrier damage in IPEC-1 cells. (**A**) The gene expression levels of ZO-1 and Occludin. (**B**) The protein expression levels of ZO-1 and Occludin. (**C**) The gene expression levels of ZO-1 and Occludin after siRNA treatment. (**D**) The protein expression levels of ZO-1 and Occludin after siRNA treatment. Data are shown as mean ± standard error representation from three independent experiments. ** p* < 0.05, *** p* < 0.01, **** p* < 0.001.

**Figure 4 antioxidants-13-00533-f004:**
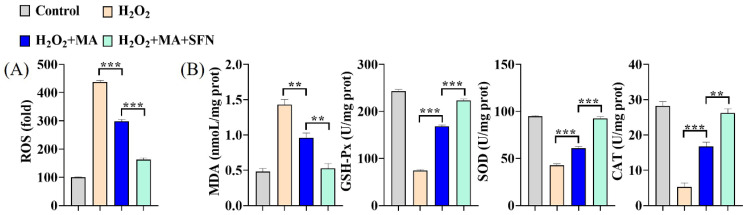
MAs attenuate H_2_O_2_-induced oxidative stress in IPEC-1 cells. The cells were first exposed to 10 μM SFN for 1 h and/or 100 mg/L of MA for 12 h, followed by stimulation with 1 mM H_2_O_2_ for 1 h. (**A**) The intracellular ROS levels were assessed by quantifying DCF fluorescence employing an enzyme marker. (**B**) The concentration of MDA was assessed using 2-thiobarbituric, while the activities of antioxidant enzymes such as SOD, GSH-Px, and CAT were measured utilizing ELISA kits. *** p* < 0.01, **** p* < 0.001.

**Figure 5 antioxidants-13-00533-f005:**
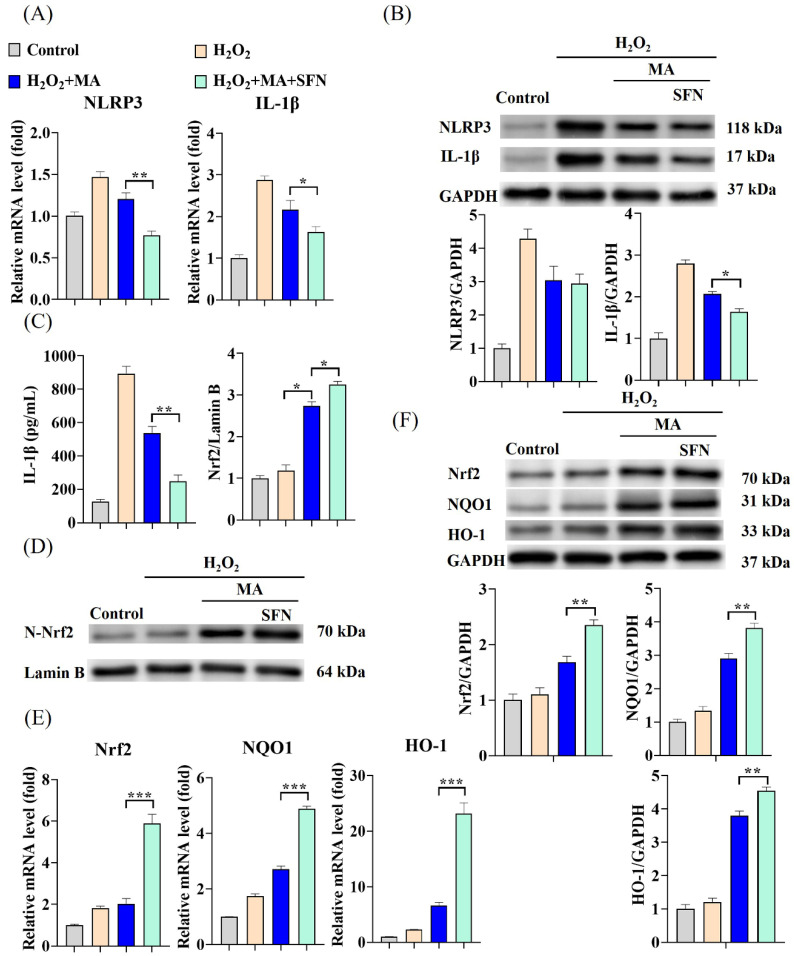
MAs alleviate H_2_O_2_-induced cellular inflammation via the Nrf2/NLRP3/IL-1β signaling pathway. The cells were first exposed to 10 μM SFN for 1 h and/or 100 mg/L of MAs for 12 h, followed by stimulation with 1 mM H_2_O_2_ for 1 h. (**A**) The gene expression levels of NLRP3 and IL-1β. (**B**) The protein expression levels of NLRP3 and IL-1β. (**C**) The protein level of IL-1β. (**D**) The protein expression level of N-Nrf2 (Nuclear-Nrf2). (**E**) The gene expression levels of Nrf2, NQO1 and HO-1. (**F**) The protein expression levels of Nrf2, NQO1, and HO-1. ** p* < 0.05, *** p* < 0.01, **** p* < 0.001.

**Figure 6 antioxidants-13-00533-f006:**
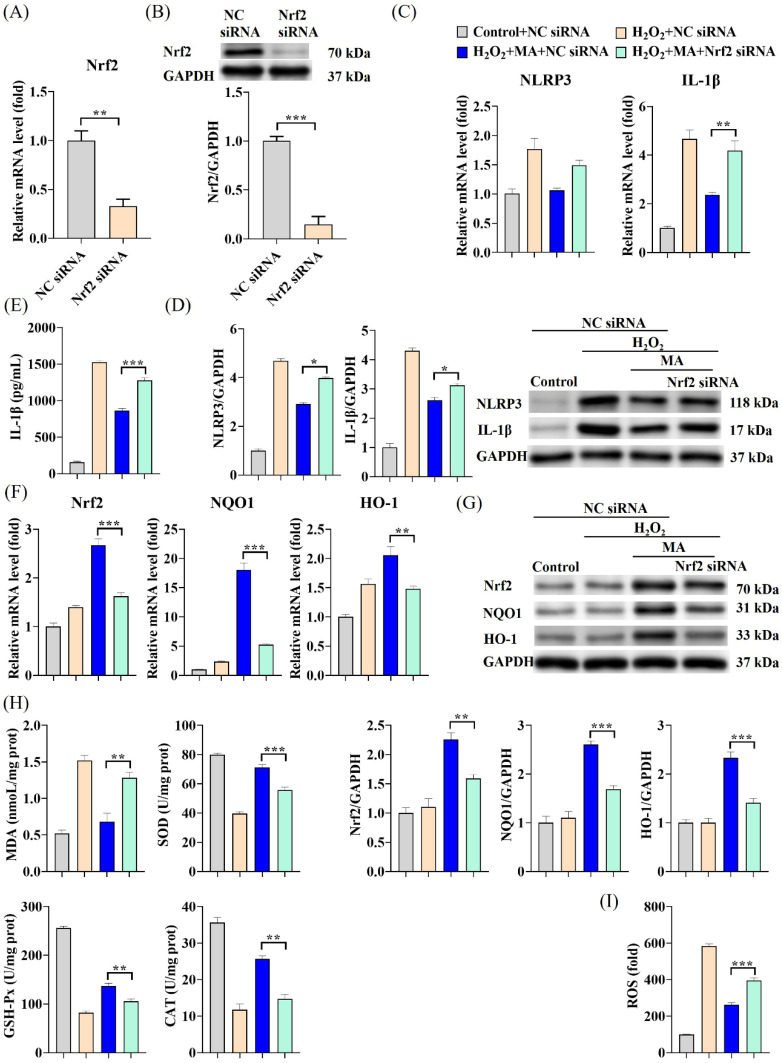
MAs alleviate H_2_O_2_-induced cellular inflammation via the Nrf2/NLRP3/IL-1β signaling pathway. The cells underwent a sequential process: they were initially transfected with siRNA for 48 h, then exposed to 100 mg/L of MAs for 12 h, and subsequently subjected to treatment with 1 mM of H_2_O_2_ for 1 h. (**A**) The gene expression level of Nrf2 after siRNA treatment. (**B**) The protein expression level of Nrf2 after siRNA treatment. (**C**) The gene expression levels of NLRP3 and IL-1β after siRNA treatment. (**D**) The protein expression levels of NLRP3 and IL-1β after siRNA treatment. (**E**) The protein level of IL-1β after siRNA treatment. (**F**) The gene expression levels of Nrf2, HO-1, and NQO1 after siRNA treatment. (**G**) The protein expression levels of Nrf2, HO-1, and NQO1 after siRNA treatment. (**H**) The concentration of MDA was assessed using 2-thiobarbituric, while the activities of antioxidant enzymes such as SOD, GSH-Px, and CAT were measured utilizing ELISA kits. (**I**) The intracellular ROS levels were assessed by quantifying DCF fluorescence employing an enzyme marker. ** p* < 0.05, *** p* < 0.01, **** p* < 0.001.

**Figure 7 antioxidants-13-00533-f007:**
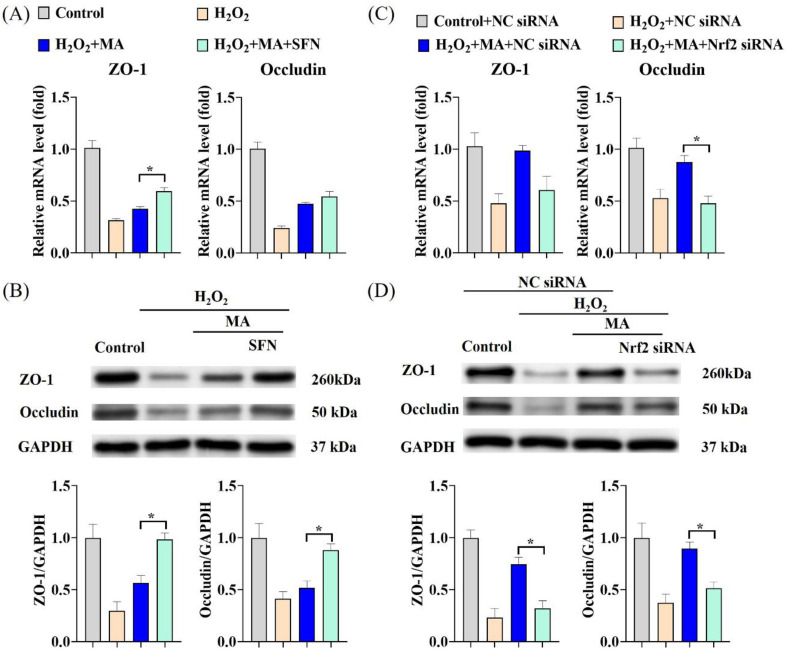
MAs attenuate H_2_O_2_-induced barrier damage by activating the Nrf2/NLRP3/IL-1β signaling pathway. (**A**) The gene expression levels of ZO-1 and Occludin with SFN treatment. (**B**) The protein expression levels of ZO-1 and Occludin with SFN treatment. (**C**) The gene expression levels of ZO-1 and Occludin after siRNA treatment. (**D**) The protein expression levels of ZO-1 and Occludin after siRNA treatment. * *p* < 0.05.

**Figure 8 antioxidants-13-00533-f008:**
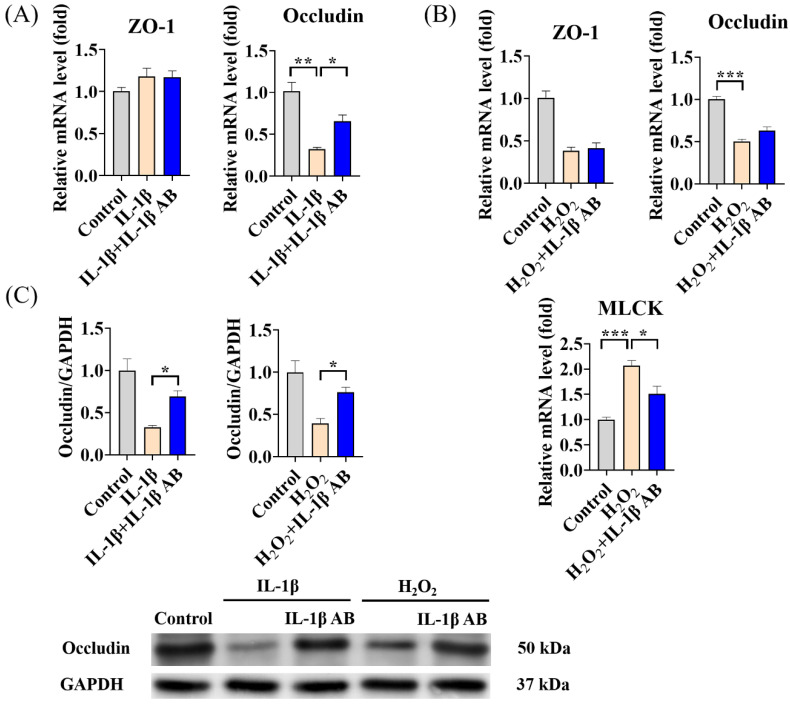
MAs attenuate H_2_O_2_-induced intestinal epithelial tight junction protein disruption by reducing IL-1β. The cells were treated with 10 ng/mL IL-1β and/or 100 ng/mL IL-1β mAb for 48 h. (**A**) The gene expression levels of ZO-1 and Occludin with IL-1β/IL-1β AB treatment. (**B**) The gene expression levels of ZO-1, Occludin and MLCK with H_2_O_2_/IL-1β AB treatment. The cells were treated with 1 mM H_2_O_2_ and/or 100 ng/mL IL-1β mAb for 1 h. (**C**) The protein expression levels of Occludin with IL-1β/IL-1β AB and H_2_O_2_/IL-1β AB treatment. * *p* < 0.05, ** *p* < 0.01, *** *p* < 0.001.

**Figure 9 antioxidants-13-00533-f009:**
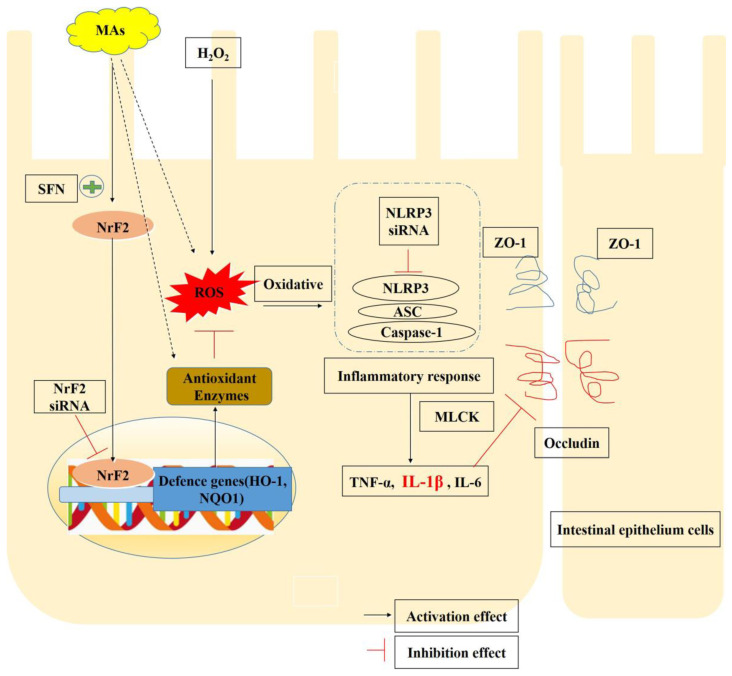
MAs protect IPEC-1 cells from H_2_O_2_-induced oxidative stress, inflammation and tight junction protein disruption via activating the Nrf2 pathway to inhibit the ROS/NLRP3/IL-1β signaling pathway. The dashed lines represent conjectures derived from prior studies.

## Data Availability

Data is contained within the article and Appendix A.

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
