# Peer review of "Microbe-Derived Antioxidants Protect IPEC-1 Cells from H2O2-Induced Oxidative Stress, Inflammation and Tight Junction Protein Disruption via Activating the Nrf2 Pathway to Inhibit the ROS/NLRP3/IL-1β Signaling Pathway"

_antioxidants, 2024, doi:10.3390/antiox13050533_

Round 1

Reviewer 1 Report

Dear Editor, Dear Authors,

I was invited to evaluate the manuscript

In this study, the authors investigated the ability of intestinal microbial products to prevent cell oxidation caused by H2O2 in intestinal cells, using pig IPEC-1 cells. To evaluate the protection, authors used a pannel of markers linked to oxidation and inflammation. Reported data show that MA possess protective effects related to the NLRP3 inflammasome as demonstrated by the use of siRNA. The authors observed an upregulation of Nrf2, NQO1, and HO-1 expression, with a parallel activation of the Nrf2-HO-1 signaling pathway, increase in the activities of antioxidant enzymes like SOD, GSH-Px, and CAT, a decrease in the accumulation of ROS and inflammation. siRNA knockdown of the Nrf2 gene demonstrates the role of this factor in MA protective effect. Overall, all data suggest that MA protect intestinal cells from H2O2-triggered oxidative stress, inflammation, and disruption of intestinal epithelial tight junctions through the activation of the Nrf2 pathway.

Overall, I found the study well designed and well conducted.

Please find below my comments.

1- Although apparently already published, please provide details on MA production and MA compositons in the Materials and Methods section

2- When studying the tight junctions, it is expected to see transepithelial electrical resistance (TEER) data as a reflect of TJs functionnality. Please measure TEER as it is a very simple assay, and does not cost so much (compared to what the authors already did in term of analysis) : just need to buy EVOM ohmmeter. If not, variations of mRNA/proteins level of TJs proteins is not enough.

3- Please check the manuscript for « increase/decrease of » to be changed to « increase/decrease in »

regards

Dear Editor, Dear Authors,

I was invited to evaluate the manuscript

In this study, the authors investigated the ability of intestinal microbial products to prevent cell oxidation caused by H2O2 in intestinal cells, using pig IPEC-1 cells. To evaluate the protection, authors used a pannel of markers linked to oxidation and inflammation. Reported data show that MA possess protective effects related to the NLRP3 inflammasome as demonstrated by the use of siRNA. The authors observed an upregulation of Nrf2, NQO1, and HO-1 expression, with a parallel activation of the Nrf2-HO-1 signaling pathway, increase in the activities of antioxidant enzymes like SOD, GSH-Px, and CAT, a decrease in the accumulation of ROS and inflammation. siRNA knockdown of the Nrf2 gene demonstrates the role of this factor in MA protective effect. Overall, all data suggest that MA protect intestinal cells from H2O2-triggered oxidative stress, inflammation, and disruption of intestinal epithelial tight junctions through the activation of the Nrf2 pathway.

Overall, I found the study well designed and well conducted.

Please find below my comments.

1- Although apparently already published, please provide details on MA production and MA compositons in the Materials and Methods section

2- When studying the tight junctions, it is expected to see transepithelial electrical resistance (TEER) data as a reflect of TJs functionnality. Please measure TEER as it is a very simple assay, and does not cost so much (compared to what the authors already did in term of analysis) : just need to buy EVOM ohmmeter. If not, variations of mRNA/proteins level of TJs proteins is not enough.

3- Please check the manuscript for « increase/decrease of » to be changed to « increase/decrease in »

regards

Reviewer 2 Report

Design of the study and presentation: The authors must provide data (e.g. Figure S1), showing the mRNA and protein levels of analyzed mRNA and protein markers in cells pre-treated with 100 mg/L MA. This is an important control, as MA alone might alter the levels of mRNA for TNF-alpha, IL-1beta, IL-6, IN18, NLRP3, ASC, Caspase-1, ZO-1 and Occludin, as well as the protein levels of IL-1beta, IL-18, NLRP3, ASC, Caspase-1, ZO-1 and Occludin. If the levels are not changed after pre-treatment of IPEC-1 cells with MA, this should be shown as well. Without understanding the effects of MA on IPEC-1 cells regarding the analyzed mRNA and protein markers, the overall interpretation of findings might be misleading.

The authors might consider softening the conclusion regarding the potential of MA to protect the disruption of intestinal epithelial tight junctions, as they only investigated changes in the protein expression levels of Occludin and ZO-1. However, the study does not provide any direct evidence that epithelial tight junctions will be protected by MA in vivo.

Lines 114-125: Cell Viability Assay: Please review the paragraph. In my opinion, the order of sentences appears to be incorrect. The sentence “MA was dissolved in a complete medium, … should proceed the sentence “To assess the impact of various concentrations of MA on …”. It should also be clearly indicated here, as well as in other places, if the H2O2 was diluted in complete medium and then applied to DPBS washed cells. Additionally, there is a repetition of word “incubated” in line 123.

Lines 129-130: Measurement of Intracellular ROS: Please specify whether the medium containing 1 mM H2O2 was replaced with fresh culture medium before adding the DCFH-DA probe.

Lines 216 and 239: It should not be referred to as “combined treatment using MA”, but rather as “pre-treatment with MA”. Please correct this.

Line 250: It should be indicated here that SFN mentioned in Figure 4 is the Nrf2 activator.

Line 343: The reference regarding the findings discussed in lines 340-343 should be included here.

Reviewer 3 Report

In their manuscript entitled "Microbe-Derived Antioxidants Protect IPEC-1 Cells from H2O2-Induced Oxidative Stress, Inflammation and Tight Junction Disruption via Activating the Nrf2 Pathway to Inhibit the ROS/NLRP3/IL-1β Signaling Pathway" the authors present a very interesting research with fundamental new findings.

Statistical analysis is carried out very well, research design was very well choosen.

The new  findings are very well, new, contribute to the advances in this field and the manuscript is suitable for publication after some revisions (see details below).

2.2. Cell Culture and Processing :
- how many passages were perfomed ?

2.3. Cell Viability Assay:

- cultered to obtain confluent layer ?

2.5. Determination of Antioxidant Enzyme Activity:
- please provide some more detail

In general: please give some more experimental detail, so experimental could be followed easier.

Please provide a section "Conclusions" after the Discussion section (line 408) ; please give also limitions of your study (or can it be generalized without any limitions ?).

IPEC-1 are porcine cells, so is there a direct transmission of the results on Human possible ? Please explain and give some limtations

Round 2

Reviewer 2 Report

I must congratulate the authors on their responses to my suggestions and comments. The revised version of the manuscript is better. 

One last comment: please mention Figure S1: "Effect of MA treatment on NLRP3, IL-1β, ZO-1 and Occludin mRNA expression" in the revised version of the manuscript. Please ensure that Figure S1 is included in the supplementary data.

-
